

# Relationship between iron status markers and insulin resistance: an exploratory study in subjects with excess body weight

M. Pilar Vaquero[1], Daniel Martínez-Maqueda[1,2], Angélica Gallego-Narbón[1,3], Belén Zapatera[1] and Jara Pérez-Jiménez[1]

[1] Department of Metabolism and Nutrition, Institute of Food Science and Technology and Nutrition (ICTAN-CSIC), Madrid, Spain
[2] Madrid Institute for Rural, Agricultural and Food Research and Development (IMIDRA), Madrid, Spain
[3] Department of Biology, Universidad Autónoma de Madrid (UAM), Madrid, Spain, España

## ABSTRACT

**Background.** Controversy exists on the relationship between iron metabolism and cardiometabolic risk. The aim of this study was to determine if there is a link between dysmetabolic iron and cardiometabolic markers in subjects with excess body weight.
**Methods.** Cross-sectional study with fifty participants presenting overweight or obesity and at least another metabolic syndrome factor. Determinations: anthropometry, body composition, blood pressure, lipids, glucose, insulin, leptin, areas under the curve (AUC) for glucose and insulin after an oral glucose tolerance test, hs-C reactive protein (hs-CRP), blood count, ferritin, transferrin, transferrin saturation (TSAT), soluble transferrin receptor (sTfR). Gender-adjusted linear correlations and two independent samples $t$ tests were used.
**Results.** Ferritin was positively correlated with insulin-AUC ($r = 0.547$, $p = 0.008$) and TSAT was negatively correlated with waist-hip ratio ($r = -0.385$, $p = 0.008$), insulin ($r = -0.551$, $p < 0.001$), and insulin resistance (HOMA-IR, $r = -0.586$, $p < 0.001$). Subjects with TSAT $\leq 20\%$ had higher insulin ($p = 0.012$) and HOMA-IR ($p = 0.003$) compared to those with TSAT $> 20\%$. In conclusion, the observed results suggest that iron transport and storage are altered in subjects with overweight/obesity, at the same time that they exhibit the characteristic features of insulin resistance. Nevertheless, this occurs without iron overload or deficiency. These results should be validated in wider cohorts since they suggest that iron transport and storage should be assessed when performing the clinical evaluation of subjects with excess body weight.

# INTRODUCTION

The worldwide epidemic of obesity that affects both industrial and developing countries (*NCD Risk Factor Collaboration, 2016*) clearly has a multifactorial origin, therefore requiring a multidisciplinary approach. Overweight and obesity are the result of an imbalance between energy intake and expenditure, but they also involve a complex

Corresponding author
Jara Pérez-Jiménez,
jara.perez@ictan.csic.es

phenomenon of endocrine disruption signals. Obesity is one of the defining factors of metabolic syndrome (MetS), which is characterised by a cluster of factors also including excessive abdominal fat, dyslipaemia, hypertension, and insulin resistance (*Alberti et al., 2009*).

Within this context, different studies have detected iron deficiency in obesity; this has been mainly attributed to the inflammatory status characteristic of excessive adipose tissue involving an elevation of proinflammatory cytokines such as hepcidin, which reduces iron absorption (*Tussing-Humphreys et al., 2009*; *Aigner, Feldman & Datz, 2014*; *Zhao et al., 2015*). In contrast, other reports show moderate to high levels of body iron in patients with cardiovascular diseases and type 2 diabetes (T2DM) (*Zacharski et al., 2017*), as well as associations between high ferritin and high triglycerides with low HDL-cholesterol in T2DM and patients with type 1 diabetes (*Vaquero et al., 2017*). Indeed, since iron activates many oxidative processes, excessive free iron may contribute either to the onset or to the aggravation of these pathologies. Therefore, if excess of iron is a contributing factor to cardiometabolic alterations, a reduction of its body levels should be beneficial. In this line, a reduction of body iron stores induced by phlebotomies decreased blood pressure and markers of cardiovascular risk, also improving glycaemic control, in patients with MetS (*Houschyar et al., 2012*). However, in patients with iron overload the benefit of this procedure compared to dietary and lifestyle intervention has not been proved (*Laine et al., 2017*). On the contrary, if iron deficiency coexists with obesity and contributes to fatigue, low physical performance, etc., iron supplementation should be promoted, but this has not been demonstrated and the issue is controversial. Another important aspect is that iron is a very well-known factor that increases food intake, which may increase body weight and adiposity in predisposed individuals, and, in this regard, one study suggested a relationship between adipocyte iron and appetite mediated by leptin, the satiety hormone (*Gao et al., 2015*).

Therefore, the complex metabolic alterations implicated in the MetS, with obesity as a key initial factor, may include dysmetabolic iron, although the specific direction of this alteration—excess or deficit—is not completely known and the role of iron in subjects with MetS, obesity, or diabetes is still limited and not conclusive.

Thus, the aim of this exploratory study was to evaluate the relationship between iron status parameters and both cardiometabolic risk factors as insulin resistance indexes in adults with overweight or obesity in order to advance in the knowledge of iron involvement in cardiometabolic alterations.

## MATERIALS & METHODS

### Subjects and determinations

This human study was conducted according to the Declaration of Helsinki guidelines. The participants were recruited in Madrid (Spain) and signed a written informed consent approved by the Ethics Committees of Hospital Puerta de Hierro-Majadahonda (2016/12/02) and the Spanish CSIC (2016/12/13). The design was a cross-sectional analytical study, part of the registered clinical trial NCT03076463. Inclusion criteria

were: age $\geq 18$ and $<70$ years, apparently healthy and presenting body mass index (BMI) $>25\,kg/m^2$ and at least another MetS factor (fasting glucose $\geq 100$ mg/dL; HDL-cholesterol $\leq 50$ mg/dL in women and 40 mg/dL in men; triglycerides $\geq 150$ mg/dL; systolic pressure $\geq 130$ mmHg or diastolic pressure $\geq 85$ mmHg). Exclusion criteria were: diagnosis of any cardiometabolic pathologies, e.g., T2DM; any previous cardiometabolic event, e.g., heart stroke; any medication related to cardiometabolic pathologies or risk factors, e.g., statins or angiotensin-converting enzyme inhibitors; any medication that may affect lipid profile; pregnancy; lactation.

Sample size was calculated using the G*Power software for bivariate normal models. The primary outcome variable for sample size calculation was the modification in the HOMA-IR (homeostatic model assessment for insulin resistance) index. Based on previous studies (*Vaquero et al., 2017*; *Ebron et al., 2015*), assuming a coefficient $r = 0.40$, $\alpha = 0.05$ and statistical power $(1 - \beta) = 0.80$, a sample size of 46 was needed. Fifty subjects were recruited, and finally 49 subjects participated. Volunteers attended the Human Nutrition Unit of our institute for fasting analyses and, due to logistic reasons, an oral glucose tolerance test (OGTT) was performed on 25 randomly selected subjects.

Blood pressure, body weight, height, waist-hip ratio (WHR), and waist to height ratio (WHtR) were measured by standardized procedures and BMI was calculated. Body composition was estimated by bioimpedance (Tanita BC-601, Arlington heights, IL, USA). Blood samples were collected to determine: haematocrit, mean corpuscular volume (MCV), haemoglobin, red blood cell distribution width (RDW), serum iron, serum transferrin, glucose, lipids, high sensitive C-reactive protein (hs-CRP), and serum ferritin, all by autoanalyzer. Serum soluble transferrin receptor (sTfR) was determined by ELISA (DRG Instruments GmbH, Marburg, Germany).

Food intake was assessed by three 24 h dietary recalls performed by telephone interviews (two in working days and one in the weekend) several days before blood sampling. They were performed by a dietitian, for about 10 min, the morning after the day that was registered. In order not to originate a modification in dietary habits, the subjects did not know in advance when they would receive the call. No diet record diaries had to be filled by volunteers. Daily energy intake, nutrient intake and energy provided by macronutrients were calculated with the computer program DIAL (Alce Ingeniería) using the Spanish Food Composition Database (*Ortega et al., 2013*).

An oral glucose tolerance test (OGTT) using a 75 g glucose solution was performed in a subsample of 25 volunteers randomly selected. Serum samples were collected at 0, 30, 60 and 120 min. Glucose was analysed by the Free Style Optimum Neo blood glucose meter (Abbott, Chicago, IL, USA) and insulin and leptin by ELISA assays (Merck-Millipore, Burlington, MS, USA). In the case of the insulin kit (EZHI-14K), the standard curve range was 2–200 µU/mL and the limit of detection was 1 µU/mL. In the case of the leptin kit (EZHL-80SK), the standard curve range was 0.5–100 ng/mL and the limit of detection was 0.2 ng/mL.

## Calculations

Total iron binding capacity (TIBC) $= 25.1 \times$ transferrin (g/L).

Transferrin saturation (TSAT) = serum iron (μmol/L)/TIBC (μmol/L) × 100.

Insulin resistance was estimated by HOMA-IR = (glucose, mg/dL × insulin, μU/mL) /405.

Areas under the curve (AUC) of glucose and insulin were calculated from the values obtained at 0, 30, 60 and 120 min of the OGTT test using the trapezoid method.

## Statistical analyses

The distribution of the variables was tested by the Shapiro–Wilk test. The relationship between variables was firstly explored by partial Pearson's correlations controlled by gender, using log- or square root-transformed variables when necessary, and $p$-values $\leq 0.01$ were considered significant. Two samples $t$ test, or Mann–Whitney test for non-normally distributed variables, were used to compare groups of subjects classified according to gender and the TSAT cut-off value of 20% indicative of iron deficiency (Wish, 2006) and differences with $p < 0.05$ were considered significant. Statistical analyses were performed with SPSS 24 (SPSS Inc., Chicago).

## RESULTS

All volunteers presented overweight or obesity, 65% high blood pressure, 10% high glucose and 22% showed both MetS factors, without significant differences between genders. WHR, haematocrit, ferritin, sTfR, and the hepatic enzymes were significantly higher in men, while body fat, abdominal fat, transferrin, leptin and HDL-cholesterol were significantly higher in women (Table 1). Concerning dietary intake (mean of the three dietary recalls), men ingested significantly more energy, protein, lipids, cholesterol and calcium than women (Table 2), although values were rather low for overweight/obese individuals.

Based on these cardiometabolic parameters, several correlations were assessed. BMI was positively correlated with body fat ($r = 0.859$, $p < 0.001$), abdominal fat ($r = 0.804$, $p < 0.001$), and HOMA-IR ($r = 0.395$, $p < 0.006$). Transferrin and TSAT were negatively correlated ($r = -0.450$, $p = 0.002$) but there were no correlations between ferritin and TSAT, ferritin and sTfR, or transferrin and sTfR. In addition, ferritin was positively correlated with insulin-AUC ($r = 0.547$, $p = 0.008$) while TSAT was negatively associated with WHR ($r = -0.385$, $p = 0.008$), insulin ($r = -0.551$, $p < 0.001$), and HOMA-IR ($r = -0.586$, $p < 0.001$). The relationship between ferritin and hs-CRP or between other iron and cardiometabolic markers did not reach the significance level ($p > 0.01$).

Additionally, the differences in cardiometabolic markers between subjects classified according to the TSAT cut-off value of 20% were evaluated (Table 3). Subjects in the low TSAT group (mean ± SD, 15.8 ± 3.4%) presented significantly lower serum iron and significantly higher RDW, transferrin, TIBC, insulin, and HOMA-IR, and marginally higher hs-CRP. The differences in HOMA-IR between these groups remained when men and women were studied separately.

## DISCUSSION

In this exploratory study, the relationship between iron metabolism and cardiometabolic markers was evaluated in adults with overweight or obesity presenting at least another

**Table 1 Body composition, blood pressure, haematological, and biochemical markers of the participants.**

|  | Men | Women | *P* value |
|---|---|---|---|
| General characteristics |  |  |  |
| Number of subjects | 27 | 22 | – |
| Age (years) | 42.5 ± 10.8 | 42.7 ± 12.5 | 0.942 |
| BMI (Kg/m$^2$) | 29.8 (32.0;27.9) | 29.2 (35.5;26.3) | 1.000 |
| WHR | 0.99 ± 0.05 | 0.88 ± 0.05 | **<0.001** |
| WHtR | 0.60 ± 0.07 | 0.61 ± 0.08 | 0.715 |
| Total body fat (%) | 27.0 ± 6.0 | 39.8 ± 7.5 | **<0.001** |
| Abdominal fat (%) | 29.3 ± 6.5 | 36.9 ± 8.9 | **0.001** |
| Systolic pressure (mmHg) | 120.8 ± 11.1 | 115.1 ± 14.7 | 0.144 |
| Diastolic pressure (mmHg) | 83.2 ± 7.8 | 81.2 ± 10.1 | 0.441 |
| Haematological parameters |  |  |  |
| Haemoglobin (g/dL) | 15.7 ± 1.0 | 14.0 ± 0.9 | **<0.001** |
| Haematocrit (%) | 47.0 (48.2;45.5) | 43.0 (44.4;40.4) | **<0.001** |
| MCV (fL) | 90.0 ± 3.8 | 90.8 ± 5.9 | 0.593 |
| RDW (%) | 13.5 ± 0.9 | 13.2 ± 0.6 | 0.169 |
| Iron ($\mu$mol/L) | 15.0 ± 5.1 | 16.6 ± 6.2 | 0.328 |
| Ferritin (ng/mL) | 160 (224.5;89.0) | 34.7 (80.6;24.2) | **<0.001** |
| Transferrin (g/L) | 2.8 ± 0.3 | 3.1 ± 0.4 | **0.014** |
| TSAT (%) | 22.8 ± 5.3 | 23.0 ± 7.4 | 0.883 |
| TIBC ($\mu$mol/L) | 70.3 ± 7.5 | 76.8 ± 9.9 | **0.014** |
| sTfR ($\mu$g/mL) | 0.99 (1.15;0.88) | 0.81 (0.98;0.68) | **0.015** |
| Cardiometabolic markers |  |  |  |
| Glucose (mg/dL) | 99.1 ± 9.1 | 98.0 ± 12.0 | 0.735 |
| Insulin ($\mu$U/mL) | 7.0 ± 2.3 | 7.4 ± 3.7 | 0.635 |
| Leptin (ng/mL) | 13.5 (28.9;7.2) | 33.6 (48.6;26.2) | **0.002** |
| Total cholesterol (mg/dL) | 199.1 ± 31.9 | 208.6 ± 53.5 | 0.442 |
| LdL-cholesterol (mg/dL) | 123.4 ± 30.0 | 122.3 ± 30.0 | 0.898 |
| HdL-cholesterol (mg/dL) | 44.1 ± 9.6 | 53.8 ± 11.0 | **0.002** |
| Triglycerides (mg/dL) | 135.5 (210.5;106) | 119.5 (156.5;85.0) | 0.183 |
| hs-CRP (mg/L) | 1.34 (3.55;0.62) | 3.32(6.60;0.70) | 0.117 |
| HOMA-IR | 1.62 (2.07;1.36) | 1.58 (2.11;1.15) | 0.355 |

**Notes.**

Values are shown as mean ± SD for normal variables, and as median (percentile 75; percentile 25) for non-normal variables. BMI, Body Mass Index; WHR, Waist to Hip Ratio; WHtR, Waist to Height Ratio; MCV, Mean Corpuscular Volume; RDW, Red blood cell Distribution Width; TSAT, Transferrin Saturation; TIBC, Total Iron Binding Capacity; sTfR, Soluble Serum Transferrin Receptor; hs-CRP, high sensitivity-C Reactive Protein; HOMA-IR, Homeostasis Model Assessment Index.

MetS factor. Results, obtained for the whole group as well as for the two groups classified according to the TSAT cut-off point, show a link between insulin resistance and iron storage, with low iron transport efficiency.

There is an intense debate on the possibility that iron dysregulation may be involved in insulin resistance. Population studies have reported associations between ferritin and cardiovascular markers, concluding that high body iron may be implicated in cardiovascular

**Table 2** Diet characteristics of the participants of the study.

|  | Men | Women | P value[a] |
|---|---|---|---|
| Energy (Kcal) | 2055 ± 565 | 1674 ± 620 | **0.037** |
| Protein (g) | 89.4 ± 17.9 | 68.3 ± 26.3 | **0.003** |
| Carbohydrates (g) | 196.4 ± 58.8 | 178.0 ± 77.0 | 0.368 |
| Fibre (g) | 21.7 (32.4;16.0) | 18.3 (29.1;14.8) | 0.328 |
| Lipids (g) | 91.4 ± 32.3 | 69.4 ± 30.6 | **0.024** |
| Cholesterol (mg) | 319.6 ± 101.7 | 253.7 ± 114.8 | **0.047** |
| SFA[b] (% energy) | 13.0 ± 2.7 | 12.5 ± 2.8 | 0.568 |
| MUFA[c] (% energy) | 16.8 ± 4.3 | 14.7 ± 5.0 | 0.159 |
| PUFA[d] (% energy) | 5.8 ± 1.9 | 5.9 ± 1.6 | 0.811 |
| Calcium (mg) | 906.0 ± 340.0 | 625.5 ± 262.4 | **0.004** |
| Iron (mg) | 14.8 ± 4.7 | 12.5 ± 4.7 | 0.105 |
| Vitamin A ($\mu$g) | 683 (1210;573) | 568 (797;424) | 0.133 |
| Vitamin B$_1$ (mg) | 1.3 (1.9;1.1) | 1.2 (1.7;1.0) | 0.315 |
| Vitamin B$_2$ (mg) | 1.9 ± 0.6 | 1.5 ± 0.6 | 0.068 |
| Folic acid ($\mu$g) | 278 (343;193) | 259 (286;172) | 0.307 |
| Vitamin C (mg) | 117 (201;55) | 107 (157;73) | 0.672 |

**Notes.**

[a] Mean values for 3 days dietary recalls. Values are expressed as mean ± SD for normal variables, and as median (percentile 75; percentile 25) for non-normal variables.

[b] SFA, Saturated Fatty acids.

[c] MUFA, Monounsaturated Fatty Acids.

[d] PUFA, Polyunsaturated Fatty Acids.

and diabetes risk (*Cheung et al., 2013*; *Wlazlo et al., 2015*). However, in most cases ferritin values were in normal physiological range, what limits the relevance of the findings. Similarly, we did not observe either abnormally high or low ferritin values in this study on subjects with overweight/obesity. We found no association between this marker and body weight, in contrast with others (*Tussing-Humphreys et al., 2009*; *Aigner, Feldman & Datz, 2014*). Nevertheless, since ferritin is an acute phase-protein, it was relevant to determine whether inflammation was a confounder factor. This was done by analysing hs-CRP, but the obtained values were below the <2 mg/dL cut-off suggested for normality (*Goff Jr et al., 2014*), so no association between ferritin and hs-CRP values was observed. Therefore, our results do not support that ferritin values were increased by inflammation in these individuals with overweight/obesity, which is in agreement with other studies (*Crist et al., 2009*). Moreover, the values of sTfR, that are unaffected by inflammation and reflect tissue iron needs, were about 50% lower than those obtained in young iron-deficient women using the same analytical technique (*Blanco-Rojo et al., 2011*; *Toxqui et al., 2013*), which denotes iron sufficiency in the present work.

The 20% TSAT cut-off value has been proposed to detect iron deficiency (*Wish, 2006*). In the present study, the higher insulin and HOMA-IR values observed in the subjects with TSAT<20% are in agreement with some results of the EPIC-InterAct prospective study (*Podmore et al., 2016*), the KORA F4 study (*Huth et al., 2015*), and the US NHANES (*Cheung et al., 2013*), all of them showing associations between low TSAT and prediabetes. Other studies reported that high (*Ellervik et al., 2011*) or both low and

**Table 3** Body composition, blood pressure, biochemical markers and cardiometabolic indexes according to transferrin saturation classification.

| | Group 1 TSAT ≤20% | Group 2 TSAT>20% | P value |
|---|---|---|---|
| General characteristics | | | |
| Women (%, n) | 50.0 (8) | 42.4 (14) | 0.789 |
| Age (years) | 39.4 ± 12.8 | 44.2 ± 10.9 | 0.185 |
| BMI (Kg/m$^2$) | 30.2 (36.5;26.0) | 29.5 (32.2;26.9) | 0.714 |
| WHR | 0.95 ± 0.06 | 0.94 ± 0.09 | 0.429 |
| WHtR | 0.62 ± 0.08 | 0.60 ± 0.06 | 0.467 |
| Total body fat (%) | 32.9 ± 10.0 | 32.1 ± 8.6 | 0.794 |
| Abdominal fat (%) | 33.2 ± 9.4 | 32.2 ± 8.0 | 0.683 |
| Systolic pressure (mmHg) | 116.8 ± 10.1 | 119.1 ± 14.3 | 0.577 |
| Diastolic pressure (mmHg) | 82.4 ± 8.0 | 82.3 ± 9.6 | 0.964 |
| Haematological parameters | | | |
| Haemoglobin (g/dL) | 14.5 ± 1.6 | 15.2 ± 1.1 | 0.087 |
| Haematocrit (%) | 45.0 (47.0;39.9) | 45.5 (47.7;43.6) | 0.175 |
| MCV (fL) | 89.5 ± 4.7 | 90.7 ± 4.9 | 0.442 |
| RDW (%) | 14.0 ± 0.9 | 13.1 ± 0.5 | **0.005** |
| Iron (μmol/L) | 10.8 ± 4.5 | 17.9 ± 4.5 | **<0.001** |
| Ferritin (ng/mL) | 36.4 (166.5;22.9) | 100.5 (184.3;63.6) | 0.072 |
| Transferrin (g/L) | 3.1 ± 0.4 | 2.8 ± 0.3 | **0.006** |
| TSAT (%) | 15.8 ± 3.4 | 26.1 ± 4.3 | **<0.001** |
| TIBC (μmol/L) | 78.4 ± 9.9 | 70.8 ± 7.8 | **0.006** |
| sTfR (μg/mL) | 1.02 (1.17;0.74) | 0.93 (1.0;0.73) | 0.151 |
| Cardiometabolic markers | | | |
| Glucose (mg/dL) | 100.7 ± 14.0 | 97.5 ± 8.2 | 0.437 |
| Insulin (μU/mL) | 8.9 ± 3.7 | 6.5 ± 2.3 | **0.012** |
| Leptin (ng/mL) | 22.7 (42.0;11.2) | 22.5 (33.1;9.1) | 0.852 |
| Total-cholesterol (mg/dL) | 208.2 ± 55.7 | 202.6 ± 36.1 | 0.678 |
| LdL-cholesterol (mg/dL) | 120.4 ± 27.6 | 125.4 ± 30.3 | 0.593 |
| HdL-cholesterol (mg/dL) | 46.8 ± 11.9 | 48.9 ± 11.1 | 0.554 |
| Triglycerides (mg/dL) | 123.5 (215.9;115.1) | 135.5 (182.0;79.5) | 0.471 |
| hs-CRP (g/L) | 3.65 (8.38;8.14) | 1.72 (3.33;4.22) | 0.060 |
| HOMA-IR | 2.01 (3.89;1.54) | 1.42 (1.88;1.14) | **0.003** |

**Notes.**
Values are mean ± SD except for non-parametric variables which are expressed as median (percentile 75; percentile 25). TSAT, Transferrin Saturation; BMI, Body Mass Index; WHR, Waist to Hip Ratio; WHtR, Waist to Height Ratio; MCV, Mean Corpuscular Volume; RDW, Red blood cell Distribution Width; TIBC, Total Iron Binding Capacity; sTfR, Soluble Serum Transferrin Receptor; hs-CRP, high sensitivity-C Reactive Protein; HOMA-IR, Homeostasis Model Assessment Index.

high (*Stack et al., 2014*) TSAT were associated with diabetes risk. This group also presented slightly higher hs-CRP than the other, although inflammation appears to be low. Therefore, subjects with overweight/obesity may present insufficient delivery of iron to tissues, which is related to insulin resistance and probably to a state of subclinical inflammation.

Different proteins play crucial roles in iron metabolism and are essential to guarantee iron binding for transport and storage, acting as scavengers. In this regard, transferrin

may be partly glycated (*Fernandez-Real, McClain & Manco, 2015*) and unable to efficiently bind and release iron, with possible physiological consequences. Since labile iron acts as a powerful pro-oxidant, free non-transferrin-bound iron may involve oxidative stress, which in turn would increase cardiovascular and T2DM risks (*Zacharski et al., 2017*; *Montonen et al., 2012*). Indeed, an elevation of non-transferrin-bound iron has been observed in patients with T2DM (*Lee et al., 2006*). This is consistent with the condition of insulin resistance observed in the present study. Leptin was also elevated in these subjects with overweight/obesity; although no reference values have been internationally established, the concentrations obtained here were above the cut-off values for cardiometabolic abnormalities reported in a Spanish sample of 11,000 subjects, i.e, 6.45 ng/mL for males and 23.75 ng/mL for females (*Gijón-Conde et al., 2015*). Nevertheless, this increase in leptin concentration was not related to any iron marker value. Therefore, considering the haematological, transferrin, TSAT, sTfR and ferritin results, it can be concluded that iron status was enough for haemoglobin maintenance, but transferrin was inefficient. Moreover, these iron metabolism proteins appear to reflect a situation of altered iron handling and iron displacement.

This study has several limitations. The main one is the small sample size and the cross-sectional design, which only allows obtaining relationships and positive/negative associations but not causality. All participants were of Caucasian origin. Therefore, results cannot be extrapolated to other races. However, the study also presents several remarkable strengths: the subjects included in this study were carefully screened; all MetS factors were monitored; and a variety of cardiometabolic and iron status markers were studied, including sTfR and possible confounders.

## CONCLUSIONS

Characteristics of iron deficiency or iron overload were not observed in subjects with overweight/obesity exhibiting at least one additional MetS factor. The issue of transferrin and its binding capacity implication on insulin resistance is much more controversial than that of ferritin, and our results suggest a complex metabolic dysregulation. Priority research should be focused not only on answering the question of whether body iron is increased or decreased in MetS, obesity and T2DM, but rather if dysmetabolic iron occurs with delocalization of iron, thus increasing its free radical potential and health consequences.

## ACKNOWLEDGEMENTS

Volunteers are kindly acknowledged for their participation in the study.

## Funding

This study was funded by the Spanish Ministry of Economy and Competitiveness (MINNECO-FEDER, grant: AGL2014 55102-JIN). Angélica Gallego-Narbón was funded by the Youth Employment Initiative (YEI) from the European Social Fund (ESF). Support for the publication fee was provided by the CSIC Open Access Publication Support Initiative through its Unit of Information Resources for Research (URICI). The funders had no role in study design, data collection and analysis, decision to publish, or preparation of the manuscript.

## Grant Disclosures

The following grant information was disclosed by the authors:
Spanish Ministry of Economy and Competitiveness: AGL201455102-JIN.
European Social Fund (ESF).
Unit of Information Resources for Research (URICI).

## Competing Interests

Jara Pérez-Jiménez is an Academic Editor for PeerJ.

## Author Contributions

- M. Pilar Vaquero conceived and designed the experiments, analyzed the data, authored or reviewed drafts of the paper, and approved the final draft.
- Daniel Martínez-Maqueda performed the experiments, authored or reviewed drafts of the paper, and approved the final draft.
- Angélica Gallego-Narbón performed the experiments, analyzed the data, prepared figures and/or tables, and approved the final draft.
- Belén Zapatera performed the experiments, analyzed the data, prepared figures and/or tables, and approved the final draft.
- Jara Pérez-Jiménez conceived and designed the experiments, performed the experiments, authored or reviewed drafts of the paper, and approved the final draft.

## Human Ethics

The following information was supplied relating to ethical approvals (i.e., approving body and any reference numbers):

This study was approved by the Ethics Committees of Hospital Puerta de Hierro-Majadahonda (doc 19.16, 2016/12/02) and the Spanish CSIC (2016/12/13).

## Data Availability

The raw data are available in the Supplementary Files.

## Supplemental Information

Supplemental information for this article can be found online at http://dx.doi.org/10.7717/peerj.9528#supplemental-information.

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
