# Peer review of "Relationship between iron status markers and insulin resistance: an exploratory study in subjects with excess body weight"

_PeerJ, doi:10.7717/peerj.9528_

## Round 0.1 · original submission · Major Revisions

Dear authors, please perform all changes demanded by the reviewers, especially concerning the statistics.

·

Basic reporting

English language requires editing.

Experimental design

The methods need additions as several points have been addressed imprecisely.

Validity of the findings

In conclusions, the sentence “This approach has never been explored and may be useful both in research and in clinical practice.” should be deleted or changed, there are quite many studies assessing iron metabolism in people with metabolic syndrome, insulin resistance and type 2 diabetes

Additional comments

The study is interesting. Several aspects should be addressed.
Abstract
dysmetabolic iron?
Hemogram? Better – blood count?
“In conclusion, the observed results suggest that iron transport and storage are altered in subjects with overweight/obesity in connection with the characteristic feature of insulin resistance, and occurring without iron overload or deficiency” - this sentence is not clear, it might be divided in to sentences

English requires editing
Introduction.
P. 48. “Obesity is often the primary factor of metabolic syndrome” – obesity is associated with metabolic syndrome?
p. 56 “cardiovascular and type 2 56 diabetes (T2DM) patients” – patients with cardiovascular diseases and type 2 diabetes
p. 57 “ type 1 diabetic patients”- patients with type 1 diabetes. Please use “patient first” language here and further
p. 91 m-2 – m2
p.91 - least another MetS factor – please state clearly, which factors and cut-offs did you use (exact cut-offs for waist circumference, blood pressure, HDL and triglycerides, and if usage of medications affecting HDL and tryglycerides were taken into account)
p. 91-92 – please state clearly, what were the exclusion criteria “diagnosis or medication for cardiometabolic pathologies” – this sentence is not informative;
p.98 – why only on 25 randomly selected subjects?
p. 108-109 - “Food intake was assessed by three 24 h dietary recalls performed by telephone interviews (two in 109 working days and one in the weekend) several days before blood sampling”. Please state exactly how the interviews were performed, and how much time they took. Was the interview performed in the end of each day or the next morning? Did the participants fill in diet record diaries?
p. 116 – please indicate exact kit numbers for insulin and leptin, and add data on detection limits and concentrations used for standart curves of the kits.
Results.
Table 1.
Number of men and women is not indicated.
Table 2 – are the mean values obtained from analysis of 3 day record reported? Taking into account that the participants are overweight, the total energy consumption seems underestimated (e.g. for overweight women 1600 kkal should provide weight-loss)
Table 3 – how do you explain that insulin resistance marker HOMA-IR was actually in the normal range in these people with overweight and obesity?
calculating the metabolic syndrome score (number of positive criteria) and their comparison between the groups of different iron status might add to your results.
Discussion
p. 181 “parallelism” – please find another word
p.201 . It is a pity that HbA1c was not assessed in this study as it could add data on protein glycation in study subjects
p. 206 “Leptin was also elevated in these subjects with overweight/obesity, although it was no related with iron markers”- do the authors talk about the present study? How can they say that leptin was elevated, if they do not provide a normal range for leptin (and it does not exist to my knowledge). Please clarify this point.
Conclusion
Line 216 - “all MetS factors were monitored” – there is no clear definition for factors and cut-offs monitored in the methods
The sentence “This approach has never been explored and may be useful both in research and in clinical practice.” should be deleted or changed, there are quite many studies assessing iron metabolism in people with metabolic syndrome, insulin resistance and type 2 diabetes, e.g.

Reviewer 2 ·

Basic reporting

No comment.

Experimental design

No comment.

Validity of the findings

Kolmogorov-Smirnov test is not very sensitive way to assess normality and is no longer suggested. Instead D'Agostino-Pearson and Shapiro-Wilk normality tests are recommended, one of them or both.

In all 3 tables some data where median is showned are presented not corrrectly. Values ​​in parentheses, as it follows from the description, are the range between 25% and 75% quartile, and range does mean distance from low to high value, but authors included just one nonsense number that is hard to interpret. Must be corrected.

Supplement data files didn't precise correspond to table numbers. The insulin values ​​listed in Table 3 could not be obtained from the data presented in the Supplement.

In normal distribution, the + -1SD range is assumed to include 68% of the data set, +-2SD - appr. 95%. In case of Vitamin A it does mean that the values goes in minus (mean +-2SD=902.9+-1085.8). It is complete nonsense.

In all 3 table are lot of inconsistencies. Calculations must be completely revised.

Additional comments

The manuscript must be revised to eliminate any calculation and presentation errors
In general, presented in manuscript data seems to be interesting, however despite the enough big number of investigated objects being 50 and huge parametric material gained, but with only 3 tables in the illustrative material it looks more like a short communication. A couple of representative graph figures of the acquired data would gave a better understanding of the presented results.

Annotated reviews are not available for download in order to protect the identity of reviewers who chose to remain anonymous.

---

## Round 0.2 · Minor Revisions

Please, correct the inaccuracies spotted by Reviewer 2.

·

Basic reporting

English language has been corrected, article structure is clear

Experimental design

Authors have clarified methodological aspects, the design and methods applied are now sufficiently explained

Validity of the findings

no comment

Additional comments

The authors have addressed the points indicated in my previous review. I recommend the paper for publication

Reviewer 2 ·

Basic reporting

No comment

Experimental design

No comment

Validity of the findings

In general, authors have made changes suggested, but there still are some points to be improved, corrected.

However, I would ask you to show the IQR, the dispersion of median, not as a general range, but as the lower and upper bounds, and this should be done in all tables. Otherwise, this may mislead the exact interpretation of the results.

There are some discrepancies in p-values, comparing represented results (in manuscript tables) and calculated from provided raw Supplement data.

Detailed explanation can be fined in Attachment.

Additional comments

I recommend that the authors correct these inaccuracies before publishing.

Annotated reviews are not available for download in order to protect the identity of reviewers who chose to remain anonymous.

---

## Round 0.3 · accepted · Accept

Congrats! The article is publishable in the present form.